# Questioning the Assumptions, Sustainability and Ethics of Endless Economic Growth

Haydn Washington 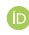

Earth and Sustainability Science Research Centre (ESSRC), School of Biological, Earth and Environmental Sciences, Biological Sciences Building (D26) Kensington Campus, UNSW, Sydney, NSW 2052, Australia; h.washington@unsw.edu.au

**Abstract:** This article questions the assumptions, sustainability and ethics of endless economic growth on the basis of environmental science, ecological economics and ecological ethics. It considers the impossibility and unsustainability of endless physical growth on a finite planet. It considers the indicators of environmental degradation (all increasing) and argues that society's addiction to endless growth is irresponsible. It discusses the key problem of denial, and how this blocks us from finding workable solutions. It discusses how in theory GDP could continue to grow modestly in the future if we adopted a steady-state economy where growth was not caused by an expanding population or resource use. However, this model is currently unpopular, with many advocating the green and circular economies that are partial solutions, and which justify ongoing growth through a fantasy of absolute decoupling. I discuss the need for society to change its anthropocentric worldview to one of ecocentrism. I then question whether the UN Sustainable Development Goals are actually ecologically sustainable. I discuss how, when we ignore the problems of an endlessly growing economy, we create significant risk to society. Rather than a focus only on 'sustainable economic growth', I suggest it is time to focus centrally on an *ecologically sustainable* economy and future.

**Keywords:** economic growth; steady-state economy; ecological economics; denial; ecocentrism; anthropocentrism; ecological ethics; decoupling; Sustainable Development Goals; ecological sustainability

## 1. Introduction

This is a Special Issue on 'Sustainable Economic Growth' in the *Journal of Risk and Financial Management* to which I was asked to submit an article. However, given the prevailing view in Western society that 'all growth is good', I will here be something of a *heretic*—as I will question the assumptions, ethics and sustainability of the premise that economic growth can (and must) increase forever. I am an environmental scientist who has written extensively on: the environmental crisis (e.g., Washington 2013, 2015, 2020a); as well as its denial in society (Washington and Cook 2011; Washington 2018); ecological economics and the steady-state economy (Washington 2014, 2017, 2020b; Washington and Twomey 2016); ecological ethics (Washington 2019, 2021; Washington et al. 2017, 2018, 2021); and the need for ecological ethics in a *new* ecological economics (Washington and Maloney 2020). My questioning of the premise that 'sustainable economic growth' is automatically *good* is thus based on environmental science, ecological economics and ecological ethics. I summarize here many issues and hence must refer readers to other sources for a detailed discussion. My purpose in this article is to take a *big picture overview* to promote discussion on what I argue is a commonly accepted dogma in the field—that all economic growth is good. We are, I feel, long overdue for a serious dialogue about the sustainability and ethics of endless growth.

## 2. Indicators of the Environmental Crisis

A discussion of sustainable economic growth must be situated within an understanding of the state of play regarding environmental degradation. Every indicator of global environmental degradation is on the rise. Washington and Kopnina (2018, p. 57) point out:

> This obsession with endless economic growth demonstrates that societies still do not understand that humanity has exceeded ecological limits, and that this is the root cause of the current environmental crisis.

Steffen et al. (2015a) in Figure 1 summarize the Earth system trends of the 'Great Acceleration' in environmental impact that has taken place since 1950. This serves to show the increasing pressure on Earth systems. This is also clearly shown in the 'Scientists' Warning to Humanity: A Second Notice' (Ripple et al. 2017) and by Crist et al. (2017).

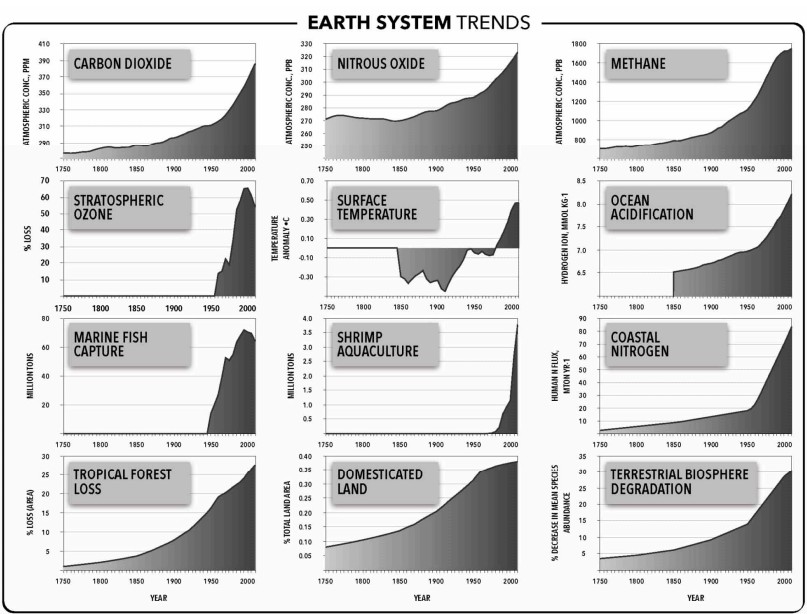

**Figure 1.** Earth System trends, adapted from the data from Steffen et al. (2015a). Note the general steep acceleration after 1950.

This has led to the following change in key environmental indicators:

○ The Global Ecological Footprint now stands at 1.7 Earths (GFN 2021). All of life on Earth of course has *only one planet*.
○ The Living Planet Index has declined by 68% since 1970 (WWF 2020).
○ The species extinction rate is at least 1000 times normal (MEA 2005). At least a million species are now threatened with extinction (IPBES 2019). Biodiversity expert E.O. Wilson (2003) predicted that if we keep proceeding this way, half the world's species will be extinct by 2100. This may in fact happen sooner (Ceballos et al. 2017).
○ At least 60% of ecosystem services are degrading or being used unsustainably (MEA 2005).
○ Three (probably now four) of nine planetary boundaries have now been exceeded as a result of human activity (Steffen et al. 2015b), as shown in Figure 2.

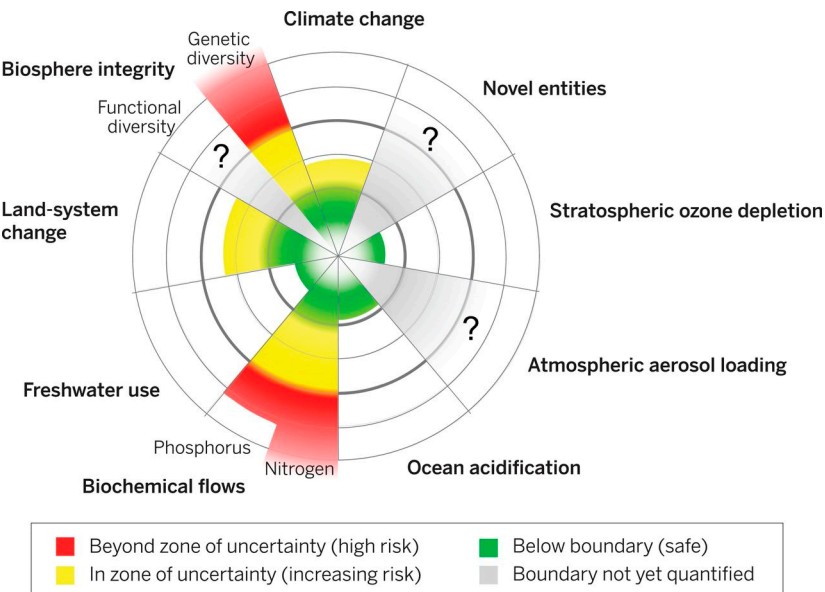

**Figure 2.** Planetary boundaries, from Steffen et al. (2015b). Note that three are at high risk and two at increasing risk.

> Washington and Kopnina (2018, p. 58) conclude:

> In effect, we are bankrupting nature and consuming the past, present and future of our biosphere (Wijkman and Rockstrom 2012). On a finite world with expanding human population and consumption, clearly something has got to give. Humanity faces a fundamental problem, for it is totally dependent on the biosphere it is degrading (Washington 2013). Hence, society needs to understand and accept that we are *way past* sustainable ecological limits.

> Discussing population has always been something of a taboo in academia (Kopnina and Washington 2016; Washington et al. 2019; Kopnina et al. 2020) but Crist et al. (2017, p. 264) conclude:

> The size of the human population is not the only variable stressing Earth. But it is a powerful force that is also eminently amenable to change, if the international political will can be mustered. Scientific willingness to engage with this issue will contribute to raising public awareness and helping to shift policies. In our efforts to halt the extinction crisis and to bequeath a biodiverse planet to future generations, willingness to marshal the resources and deploy proven tactics to address the population question is crucial.

In the past, denial of ecological limits was common in neoclassical economics (Nadeau 2009; Keen 2020; Spash and Hache 2021). However, such a denial of reality is not just a thing of the past in academia. 'An Ecomodernist Manifesto' (available at http://www.ecomodernism.org/ accessed on 17 October 2021) states that: 'there is still remarkably little evidence that human population and economic expansion will outstrip the capacity to grow food or procure critical material resources in the foreseeable future'. The recent Dasgupta (2021) review 'The Economics of Biodiversity' in the United Kingdom also continues the neoclassical denial of reality (Spash and Hache 2021).

Such a dismissal of ecological limits (and denial of the data concerning the rapidly worsening environmental crisis) indicates that many in society (and academia) remain in denial of the unsustainability of endless growth. Clearly, endless physical growth on a finite planet is unsustainable, especially when one has exceeded ecological limits (Ripple et al. 2017). The question thus becomes: 'but can the economy continue to grow if we do not grow physically?'. That question will be discussed in Section 6.

### 3. Questioning the Assumptions of Neoclassical Economics

A key tenet of neoclassical economics is endless growth (Daly 1991, 1996, 2014); however, the assumptions of neoclassical economics deserve to be closely examined. In modern industrial (aka Western) society, we are generally told that economic growth is 'always good' and should be our key goal. However, in Nature, apart from entropy, nothing grows forever. As Daly (1991) notes, the verb 'to grow' has become twisted; we have forgotten its original meaning: to spring up and 'develop to maturity'. That is, in Nature, growth gives way to maturity, a steady state. To grow beyond a certain point can be disastrous; indeed, the environmental indicators above show that this has clearly been the case (Washington and Kopnina 2018). Drawing on the work of the ecological economist Herman Daly (1991, 1996, 2014) over several decades, Washington (2014) summarized the assumptions of neoclassical economics as follows:

(1)　Strong *anthropocentrism*. Nature is seen as 'just a resource' (Washington et al. 2021).
(2)　The idea that the 'invisible hand' of the *free market* will control all that is needed for human benefit (Daly 1991). This is a key 'given truth' of neoclassical economics and has become almost a religion (Daly 2008).
(3)　The idea that the economy can *grow forever* in terms of continually rising GDP, which has increased by an astounding 25-fold over the last century (Dietz and O'Neill 2013).
(4)　The *refusal to accept any biophysical limits to growth*; for, when classical economics was developed, limits were distant (Daly 1991).
(5)　A *circular theory of production* causing consumption that causes production in a never-ending cycle. Real production and consumption, however, are in *no way circular* (Daly 1991).
(6)　Environmental damage is *merely an 'externality'*, a spillover effect of market transactions, and worth only peripheral attention (Daly and Cobb 1994).
(7)　All forms of *capital can be substituted*; thus, human capital can be substituted for natural capital (this is 'weak sustainability', Daly and Cobb 1994). This assumes that money can replace essential ecosystem services that support human society.

Assumption 1 is based on an anthropocentric worldview that asserts that Nature has no *intrinsic value* (Rolston 2012). Assumptions 2–8 are *ideological* in nature, part of a denial of reality, and they actually break the laws of thermodynamics. Assumption 1 is thus unethical, while Assumptions 2–8 are actually *irrational* as they deny physical laws and ecological limits (especially Assumptions 4 and 8) (Washington 2014). Yet, these remain the assumptions that arguably still underpin the dominant neoclassical economic synthesis today (Ibid). Such assumptions are not ecologically or ethically sustainable. Arguably, this is *why* we now have an environmental crisis.

### 4. The Impossibility and Unsustainability of Endless Physical Growth on a Finite Planet

It was in 1966 that Kenneth Boulding (1966) noted that: 'Anybody who believes in indefinite growth of anything physical, on a physically finite planet, is either mad or an economist'. Washington and Kopnina (2018, p. 57) note:

> The United Nations (UN), almost all governments, business, media and both the political 'left' and 'right' are busy extolling *endless growth*. Yet we live on a *finite* planet, so clearly endless economic growth is impossible, and its pursuit unsustainable and unethical—indeed, such destructive pursuit of the impossible is insane. There are three main drivers of 'unsustainability'—overpopulation, overconsumption and the growth economy (Washington 2015).

They further note (p. 57):

> The question "On a finite planet, is it possible to keep growing economically forever?" is one hardly ever asked in neoclassical economics (Daly 1991, 2014) or in many other academic disciplines (Washington 2015). Even the World Commission on Environment and Development (WCED 1987) report *Our Common Future*

did not ask that question—suggesting that 'sustainable development' required a gross domestic product growth rate of 5% (a rate at which the global economy would *double* its output every 14 years).

Daly (1991, p. 183) pointed out that economic growth is unrealistically held to be:

... the cure for poverty, unemployment, debt repayment, inflation, balance of payment deficits, the population explosion, crime, divorce and drug addiction.

This has not changed much in the three decades since Daly wrote those words, and economic growth is still widely seen as the panacea for almost all societal ills (e.g., Pettinger 2019). This can be seen in the Sustainable Development Goals discussed later. Sometimes, the commitment to growth may be promoted in the guise of 'free trade', 'competitiveness', 'productivity'—or even as 'sustainable development' (Victor 2008). World leaders still overwhelmingly seek growth above all else. Neoclassical economics claimed that the benefits of growth would 'trickle down' and alleviate global poverty, but this has conclusively failed (Kopnina and Blewitt 2015).

A final aspect of growthism is that it is commonly claimed that: 'economic growth is necessary if we are to have jobs'. Is this correct? There are good grounds to question whether jobs have historically been linked to growth (Washington and Kopnina 2018). Victor (2008) notes that the idea only developed 60 years ago, and for most of human history we managed to provide employment *without* major economic growth. Does growth necessarily bring employment in any case? For example, there were more Canadians with incomes less than the 'low income cut-off' in 2005 than in 1980, *despite* real Canadian gross domestic product having nearly doubled over that period (Victor 2008). As Victor (2008) explains, it is possible to develop scenarios where full employment prevails, poverty is eliminated, people have more leisure, and greenhouse gases are drastically reduced, in the context of low—and ultimately no—economic growth. It is thus arguably mistaken to assume that economic growth is a necessity for full employment.

From an environmental science viewpoint, once we have exceeded ecological limits, growth will make us worse off (Washington 2020a). We have then reached *uneconomic* growth (Daly 2014). However, unless there are changes in our social outlook, our experience of diminished well-being will be blamed by neoclassical economics (and most governments) on 'product scarcity'. The orthodox economic policy response will then be to advocate *increased growth* to remedy this. In the real world of ecological limits, this will make us even *worse* off, but this will in turn lead to advocacy of 'even more growth' (Daly 1991). Accordingly, this becomes a 'death spiral' where ever increasing growth worsens the environmental crisis. An ecologically sustainable world may require accepting the reality that the economy cannot grow forever (Washington 2020a), unless this is within the constraint of a steady-state economy (see later).

## 5. The Key Problem of Denial

Denial in society is not something we can ignore, even if there has been a silence about it until quite recently (Zerubavel 2006; Oreskes and Conway 2010; Norgaard 2011). How is it possible for civilizations to be blind towards the grave and rapidly approaching threats to their survival, even when the evidence for those threats is extensive (Brown 2008)? Several scholars point out that humanity has a key failing—we tend to *deny* our problems (Oreskes and Conway 2010; Norgaard 2011; Grusovnik 2012; Washington and Cook 2011; Washington 2018). Humanity denies some things because they force us to 'confront change', others because they are just too painful, or make us afraid. This human incapacity to hear bad news makes it hard to solve the environmental crisis (Washington and Cook 2011). However, another source of this denial is *ideological*, where the reality of the environmental crisis is denied owing to neoliberal hatred of any regulations that could restrict the activities of business (Oreskes and Conway 2010). The result of such a denial is that, as a society, we continue to act as if there is no environmental crisis, no matter what the science says (Washington 2018, 2020a; Spash and Hache 2021). This raises a key question we need to ask: 'Can ongoing economic growth be sustainable in the long-term?'.

### 6. Can the Economy Grow If We Do Not Increase the Population or Resource Use?

I am an advocate of the steady-state economy, first promoted by Herman Daly and others (Daly 1991, 1996, 2014; Dietz and O'Neill 2013; Washington and Twomey 2016). A steady-state economy is defined by three key attributes:

- A stable and ecologically sustainable population;
- A low use of resources; and
- Equitable distribution of wealth (Daly 1991; Magnus-Johnston 2016).

So, can GDP still increase under a SSE? This is a debatable point. Daly (1991) noted that it was an illusion to think that growth could continue by becoming ever less materially intensive and ever more service-oriented. Czech (2013) believes there still remains a 'fundamental conflict' between economic growth and biodiversity conservation. Winston (2012) concludes that economic growth cannot last forever. Welzer (2011) concludes that the decoupling debate maintains the illusion that we can 'just make minor adjustments'. Partington (2019) similarly asks: is it time to end our fixation with GDP growth?

In theory, GDP probably 'could' still increase in a SSE, *provided* it is not due to increasing our ecological footprint and environmental impact through increasing the human population and our use of resources (Washington 2014). Can it do so *forever,* however? Some, such as Gittins (2013) and Randers (2013), believe it could do so. However, the last 200 years have been driven by economic growth through 'more'—more people and more consumption of resources. It is these two drivers of growth we have been addicted to, and which we now need to control (Washington 2014). The GDP in our economy could grow by society doing things in a more clever, creative, innovative or sustainable manner. For example, IT can help society to do things more cleverly. The arts are endlessly creative (which can grow GDP), innovation such as renewable energy is a rapidly expanding part of the economy, and being sustainable can also convert to rising GDP (Washington 2014). I would argue this is the *only way* to sustain economic growth (albeit at a lower rate) in the long-term.

A continually rising human population is not sustainable when society is way past ecological limits (Crist et al. 2017; Ripple et al. 2017; Ceballos et al. 2017; Washington 2020a), nor is increasing resource use (IISD 2021), nor is increasing inequality in wealth (Wilkinson and Pickett 2010; Piketty 2014). So, GDP can increase through greater efficiency, appropriate technology with a lower impact, and more services and cultural activities. GDP in a steady-state economy *could* thus continue to grow, but nowhere near as fast as in the last century. Some modern economists argue in line with this, as they can see the need for change (e.g., Spash and Hache 2021). The problem is that they remain in the minority.

However, the key problem is that the SSE has been ignored or criticized by traditional neoclassical economists since it was proposed by Daly (1977). Similarly, society and governments and the United Nations have continued to deny the centrality of population (in fact *over*population) in terms of a sustainable future (Kopnina and Washington 2016; Washington et al. 2019; Kopnina et al. 2020). In addition, wealth distribution is given at best 'lip service' in a neoliberal-dominated society (Wilkinson and Pickett 2010; Piketty 2014). There has been more agreement that we must decrease the use of resources (e.g., von Weizsäcker et al. 2009) though despite this, resource use continues to grow (Cooper et al. 2018). Despite a 30% increase in resource efficiency, global resource use has expanded by 50% over 30 years (Flavin 2010). Global material resource use was expected to reach nearly 90 billion tons in 2017, and may more than double over the period from 2015 to 2050 (UNEP 2017).

*Degrowth* has become a popular term in recent years (e.g., Latouche 2010; Kallis 2018), arguing realistically that our economy is way *beyond* an ecologically sustainable level. However, one cannot degrow forever, and ideally one should degrow to a steady-state economy (Czech and Mastini 2020). At the first Degrowth Conference, its Final Declaration concluded (FICED 2008): 'once rightsizing has been achieved through the process of degrowth, the aim should be to maintain a "steady-state economy" with a relatively stable, mildly fluctuating level of consumption'. Since then, however, degrowth advocates have not talked much about a steady-state economy, largely, I argue, because the SSE refers

centrally to the issue of population. Some degrowth advocates in fact refuse to seriously discuss population (e.g., Kallis 2018). The idea of sustainable economic growth (at a modest level) via a steady-state economy is thus sadly quite 'isolated in the wilderness' of both orthodox (and unorthodox regarding degrowth) economic thought. I suggest that this is because the growth in GDP that is possible via a steady-state economy (in a sustainable way) is *far lower* than the high GDP growth rates we have become used to (and seem to assume must continue) in the last few decades.

I should note that there are other useful ecological economic models. These include Raworth's (2017) 'Doughnut economics', Spash's (2011, 2012, 2017) 'Social ecological economics' and the Sharing economy (Matofska 2016). Space precludes a detailed discussion of these here, but they have been compared by Washington and Maloney (2020). I would observe here that while all these models have merits, only the steady-state economy is strongly against growthism (and clearly foregrounds the centrality of an ecologically sustainable population).

Another important issue to briefly consider is the *Green New Deal* (GND), about which there has been much discussion (e.g., Pettifor 2019; Klein 2019; Stilwell 2020). The GND is defined by Wikipedia as being a: 'proposed United States economic stimulus package that aims to address climate change and economic inequality'. It demands total decarbonization and a commitment to an economy based on fairness and social justice. However, I would ask 'Is it truly green?'. It is entrenched in the idea that we will *grow* our way out of our problems by stimulating the economy to grow by renewable energy, green jobs and greater equality. Now, I accept that part of the idea of this is to get massive support for renewables. In this case, however, it is by using our civilization's addiction to endless growth to try and support renewables—because they will grow the economy (Washington 2020b, see Conclusion). However, given that many environmental scholars argue that climate change is in fact a symptom of society's commitment to an endlessly growing population, consumerism and economy (e.g., Rees 2008; Crist et al. 2017), maybe it is actually time to realize that the solution to get us out of this mess is to drop this endless growth idea in regard to all three? Another question to ask is 'Is it truly new?'. If justice is raised at all in regard to the GND, it is just social justice, with essentially no mention of ecological justice (Washington et al. 2018) as part of the supposedly 'new' deal? This is very much part of the deeply anthropocentric worldview of Western modernism, and as such is in no way 'new' (Curry 2011; Washington et al. 2021).

So, what has been the result of the fact that governments and academics have largely ignored the steady-state economy? Predominantly, I would argue that society has denied the need to actually move to a '*sustainable* economic growth' via a steady-state economy and has mainly argued for the mantra of 'absolute decoupling'.

## 7. The Fantasy of Absolute Decoupling

'Decoupling' refers to the idea that an economy can continue to increase its output of goods and services, *without* thereby increasing pressure on the environment—for example, by shifting to renewable energy sources, and using efficiencies to reduce the amount of resources and energy consumed (Washington and Kopnina 2018). Reducing the use of energy and materials by society is certainly needed, and some claim we can move to a 'Factor 5' strategy and only use 20% of the energy and materials we currently use (von Weizsäcker et al. 2009), whilst still retaining our current quality of life. The problem with this approach is that the very concept of decoupling suggests we can *keep on growing forever*. UNEP advocates the 'green economy', yet also sees this economy as 'a new engine of growth' (UNEP 2011: 2). This combination of 'green' and 'growth' is only made plausible by invoking the idea that it is possible to completely decouple economic growth from environmental impacts. Similarly, the circular economy supports endless growth, and rationalizes this through supporting decoupling (EMF 2014). Fletcher and Rammelt (2017, p. 450) argue that decoupling is a:

... 'fantasy' that functions to obfuscate fundamental tensions among the goals of poverty alleviation, environmental sustainability, and profitable enterprise that it is intended to reconcile. In this way, decoupling serves to sustain faith in the possibility of attaining sustainable development within the context of a neoliberal capitalist economy that necessitates continual growth to confront inherent contradictions.

They further note (p. 458) that while asserting the necessity of dramatic decoupling for any hope of genuine sustainable development, UNEP simultaneously admits that:

(1)   there is virtually no evidence that (absolute) decoupling works;
(2)   the conceptual basis for even imagining its possibility is weak; and
(3)   even if it were possible, it would be politically infeasible.

Parrique et al. (2019, p. 4) conclude:

The validity of the green growth discourse relies on the assumption of an absolute, permanent, global, large and fast enough decoupling of economic growth from all critical environmental pressures. The literature reviewed clearly shows that there is no empirical evidence for such a decoupling currently happening. (their emphasis)

Ward et al. (2016) conclude that:

It is therefore misleading to develop growth-oriented policy around the expectation that decoupling is possible. We also note that GDP is increasingly seen as a poor proxy for societal wellbeing. GDP growth is therefore a questionable societal goal.

It is worth observing that some papers that speak of 'absolute decoupling' in fact show only *relative* decoupling (e.g., Shuai et al. 2019).

How successful have we been in decoupling? Some modest decoupling of material flows occurred from the mid-1970s to the mid-1990s, but total material throughput in the global economy still increased (Flavin 2010). Victor and Jackson (2015) note that while there has been some 'relative decoupling', any serious absolute decoupling is not evident. At best, as Victor (2008) notes, attempts at decoupling *slow down* the rate at which things get worse, but do not turn them around. Perhaps the most rigorous study of 835 articles on decoupling is by Haberl et al. (2020), which concluded that:

... large rapid absolute reductions of resource use and GHG emissions cannot be achieved through observed decoupling rates, hence decoupling needs to be complemented by sufficiency-oriented strategies and strict enforcement of absolute reduction targets.

Accordingly, I believe we should be circumspect in regard to talk of 100% or absolute decoupling, as it may be merely a form of wishful thinking, one that allows 'business-as-usual' growth to continue. Indeed, focusing our attention on the idea of decoupling runs the risk of becoming part of the denial of the unsustainability of endless growth on a finite planet (Fletcher and Rammelt 2017; Washington and Kopnina 2018).

## 8. The Worldview and Ethics of Endless Growth

Many things change (and solutions become easier) if we change our worldview and ethics. As Donella Meadows (1997, p. 84) noted:

People who manage to intervene in systems at the level of a paradigm hit a leverage point that totally transforms systems [ ... ] In a single individual it can happen in a millisecond. All it takes is a click in the mind, a new way of seeing.

It has only been possible for our societies to maintain a belief in the desirability of pursuing endless growth because of the dominant anthropocentric worldview of modernism (Curry 2011; Rolston 2012; Washington 2019; Washington et al. 2021), which sees the world as no more than a 'resource' for human use (Crist 2012, 2019). To put this another way, the obsession with endless growth has been the offspring of the anthropocentric

'human chauvinism' and 'human supremacy' that has dominated Western society for at least the last 200 years (Crist 2012; Washington et al. 2017; Kopnina et al. 2018a, 2018b). In contrast, an *ecocentric* worldview finds intrinsic value in nature (Curry 2011; Washington et al. 2017). Ecocentrism upholds, as Daly (1991, p. 248) notes, that: 'there is something fundamentally wrong in treating the Earth as if it were a business in liquidation'. Society thus needs to return to *ecocentrism* and adopt an Earth ethics (Curry 2011; Rolston 2012). Changing to a worldview of ecocentrism that 'recognizes the Earth as the ultimate source of value, meaning and enablement for all beings, including—but not only—human beings' (TEC n.d.) is thus a key step on the path to a meaningful and ecologically sustainable future (Curry 2011; Rolston 2012; Washington et al. 2017, 2021).

## 9. Are the SDGs Actually Sustainable?

The UN Sustainable Development Goals (SDGs) fail to acknowledge that endless growth is impossible, and its pursuit fundamentally unsustainable (Kopnina 2016, 2020), and are arguably failing to create transformational change (Deighton 2019). The UN oversaw the development of the Sustainable Development Goals (SDGs) as discussed by Higgs (2020). The SDGs in Goal 8 state: 'Promote sustained, inclusive and sustainable economic growth'. However, despite the UN being heavily involved in studies that acknowledge that society has exceeded ecological limits (e.g., MEA 2005; Kumar 2010), Goal 8 of the SDGs continues to argue for a *sustained* growth that is portrayed as being 'sustainable'. It is also planned to be 'inclusive', though presumably (ethically) this is just for humanity, as the nonhuman world is in the process of ecological collapse (Wijkman and Rockstrom 2012; Crist et al. 2017; Ripple et al. 2017) due to the accelerating environmental crisis caused in large part by society's endless growth mantra (Daly 1991, 1996, 2014; Dietz and O'Neill 2013; Raworth 2017). Notably, the SDGs fail to discuss the key problem of overpopulation and argue quite glibly for 'no poverty' (Goal 1) and 'Zero hunger' (Goal 2). However, *how* these are to be achieved is not really stated, as an increasing population and consumption of resources is degrading the Nature that society is fundamentally reliant on (Washington 2013; Crist et al. 2017).

There are thus good grounds for questioning whether the SDGs are in fact ecologically 'sustainable' (Kopnina 2016, 2020). Many of the SDGs are praiseworthy, but are they in fact *practical* solutions when they ignore overpopulation, and support a continually growing economy? Similarly, the SDGs argue for 'Reduced inequalities' (Goal 10) when in fact the current neoclassical and neoliberal growth economy is causing increasing inequalities (Wilkinson and Pickett 2010; Piketty 2014). Horton (2014), speaking about health and sustainability issues, argues:

> The SDGs are fairy tales, dressed in the bureaucratese of intergovernmental narcissism, adorned with the robes of multilateral paralysis, and poisoned by the acid of nation-state failure.

It has also been argued in a *Nature* editorial that it is time to revise the SDGs and that one priority: 'is to decouple the SDGs from economic-growth targets' (Nature 2020).

The SDGs are thus arguably partly based on denial (rather than an acceptance) of environmental reality (Washington 2020b). This remains a major obstacle to society reaching an ecologically sustainable future when the United Nations and many governments (and academics) seek to fulfill goals that are not, in fact, *ecologically* sustainable.

## 10. Risk

This is a Special Issue of the *Journal of Risk and Financial Management*; hence, I should speak briefly about 'risk'. We know that if society denies climate change, then it places itself at great risk (Oreskes and Conway 2010; Washington and Cook 2011; Norgaard 2011). As humans are fully dependent on Nature to survive, when predictions are that we may send *half* of all species to extinction by 2100 (Wilson 2003) or sooner (Ceballos et al. 2017), then we similarly run a great risk to society's continued functioning. However, when we ignore the problems of overpopulation and overconsumption, we similarly put society at great risk

(Washington 2020a). In addition, when we ignore the problems of an endlessly growing economy (Daly 1991, 1996, 2014) on a finite planet, and seek to use what some see as denial stratagems, such as 'absolute decoupling' (Twomey and Washington 2016; Fletcher and Rammelt 2017), then we similarly create significant risk to society. History has shown us that if civilizations ignore such risks, they eventually collapse (Diamond 2005). Indeed, the risk of both ecosystem and societal collapse is growing rapidly (Ehrlich and Ehrlich 2013; Washington 2015, 2020a; Moses 2020). Despite this, much of society and academia continue to look the other way and ignore such risks, especially those associated with an endless-growth economy and an expanding human population (Washington 2020a).

I would like to reiterate what I see as a fundamental truth—that no matter how many in society continue to repeat in unison that 'all growth is good, and we can grow forever', this does not change the reality that it *is not*, and we *cannot*. I suspect some readers of this journal may disagree, but I feel that, scientifically and ethically, I should emphasize this.

## 11. Implications for Academic and Economic Sectors

One of my reviewers suggested I write a section that answers the question of: what might be the 'implications for academic and economic sectors' of this article? It is a good question. As a society, one clear implication is that there is a great need to move past denial in regard to the problems of the endless-growth economy. The economy must now operate *within ecological limits* if society is to have any chance of reaching an ecologically sustainable future. We also need to cease our addiction to endless growth in the human population, resource use, and increasing inequality. There are non-coercive solutions that can reduce and stabilize the human population (Engelman 2012), chief among which are government support for family planning, contraceptives, education for children/young adults (especially girls/young women), and banning child bride marriages (Kopnina et al. 2020). The Population Media Centre (https://www.populationmedia.org/ accessed on 17 October 2021) has been successful in creating media programs that get people to think about, and act on, family planning. We also need to reign in society's rampant consumerism (Wiedmann et al. 2020) as part of our rapid reduction in resource use. The circular economy aims to reduce resource use but fails to take a strong stance against consumerism (Twomey and Washington 2016). There will be a need to control advertising also, and Daly (2008) has suggested an advertising tax. Clearly, there will also be a need to take greater action to reduce inequality in society (e.g., https://equalitytrust.org.uk/about-us accessed on 17 October 2021). None of this will be easy, and I do accept the magnitude of the 'Great Work' (Berry 1999) of Earth repair that is needed. However, while challenging, change is *possible* (given that ecological and social collapse may be the alternative).

What of academia? Well, I am a great believer in *dialogue*—once one has moved past denial. If academia does accept that neoclassical economics is an unsustainable dogma, and that an endless-growth economy based on increasing the population and resource use is not workable, then clearly there are alternatives out there. These include the steady-state economy, degrowth, social ecological economics, doughnut economics, and even the Green New Deal. There is a possibility of *some* further growth in GDP in a steady-state economy, but it will not be like the last hundred years of rapid GDP growth; it will perforce, be modest.

That means it is time for us to recognize that rapid growth in GDP is a thing of the past. We need to get used to this. This may be staved off for some time via a Green New Deal and rapid conversion of society to 100% renewable energy (to avoid a rapidly worsening climate crisis). However, this will need to be a *transition phase*, not a phase of endless and rapid growth in GDP. Slow (or no) GDP growth will be a shock to some in business (and in government), if not perhaps to many in society who are looking for a change towards meaningful sustainability (Washington 2015; Crist 2019).

The alternative will of course be to keep denying our predicament and stay with an endless growth mantra. GDP will indeed keep growing for some time in this case—up

until the point of ecosystem and societal collapse. Given that I do not advocate for collapse, clearly I am arguing here for: (1) acceptance of the science; (2) acceptance that the economy must operate within ecological limits; (3) a commitment to ecological ethics; and (4) a willingness to *change*, to find positive steps towards an ecologically sustainable economy (Washington 2017) and future (Washington 2020a).

Can humanity do this? We have a lot of denial to break through, and a need to rapidly change our worldview and ethics (Washington et al. 2021). Will this happen—or will we remain a slave to an endless growth mantra? Whether it actually works will be, collectively, *up to all of us*.

## 12. Conclusions

The question I suggest one should really ask is whether sustainable economic growth will in fact be *ecologically* sustainable (and hence quite modest in scope), or whether society will continue to proceed in denial, and pretend that economic growth is sacrosanct, and must be prioritized and expanded above all things, including an ecologically sustainable future? Many in society (and academia) speak glibly of the 'green' and 'circular' economies, but these are largely based on the fantasy of absolute decoupling. As such, they are, I argue, part of the denial of the predicament humanity finds itself in—rapidly expanding environmental and climate crises (Washington 2020a). In terms of human survival, a key question we should ask is: 'can we keep a sustainable biosphere that can support society in the long-term?'. This is a far more important question than asking if we can keep the economy growing forever in terms of GDP. If society is going to operate via a commitment to a mantra of endless economic growth *at any cost*, then clearly the cost is already far too high, with a million species currently threatened with extinction (IPBES 2019). Our economy must operate *within ecological limits* if it is to be sustainable in the long term. It has clearly failed to do so over the last century, as neoclassical economics has largely ignored such limits.

The unsustainability of endless *physical* growth is a critical reality that society does need to acknowledge and discuss. The evidence is good that endless *economic* growth is unsustainable *if* it relies on expanding the population and resource use (as is currently the case). To ignore this is irrational and self-destructive (though admittedly society does not always act rationally). Ecological limits exist and have been hugely exceeded (Steffen et al. 2015a; Ripple et al. 2017). Yet, Western society seemingly remains locked into the unsustainable mantra of endless growth that has been a chief cause of the environmental crisis. The government and business response to this has been to undertake *partial* solutions, while at the same time denying the central cause—our addiction to endless growth. The green and circular economies are, I believe, such partial solutions, as they describe themselves as engines of *more* growth. That is not to say they are not worth doing (as part of the full solution), but they have a key weakness, as they are based on seeking endless growth, rather than on a meaningful approach to reaching ecological sustainability. By contrast to the green and circular economies, few in society and academia promote the steady-state economy, which is actually based on sustainability, and an understanding that our economy must operate *within* the ecological limits of the Earth (Daly 2014; Dietz and O'Neill 2013). Similarly, the SDGs are arguably partial solutions, as they ignore overpopulation and support the idea that economic growth *must* continue, even if they glibly label this as 'sustainable'. How such growth *can* be sustainable when one ignores the population driver of the environmental crisis is not explained (or even seriously discussed). This is clearly a denial of a major status and impact in regard to society's future.

Society's ability to deny our predicament is aided by the dominant worldview of anthropocentric modernism (Washington et al. 2021). Hence, we face a difficult predicament, for the global anthropocentric experiment of endless growth has well and truly *failed*, and destructively so (Ripple et al. 2017; Ceballos et al. 2017; Crist et al. 2017; Washington 2020a). We need to change our worldview from anthropocentrism to *ecocentrism* (Washington et al. 2017; Kopnina et al. 2018a; Taylor et al. 2020).

Change is not easy, but it *is* possible, but only by accepting the nature and scale of our predicament. If we break the silence of denial, then everything becomes much easier. We can then move to slowing (then stopping and non-coercively reducing) the human population and minimizing resource use via a steady-state economy (Daly 1991, 2014). We could accordingly stop global ecocide, improve social equality, and move to a truly ecologically sustainable future. A collapsing global ecosystem is clearly not in anybody's interests and is not going to be 'financially sustainable'. However, society could move from the egotistical 'Anthropocene' (Moore 2013) to the start of the sustainable 'Ecozoic' (Swimme and Berry 1992). A worthy vision for today is the 'Great Work' (Berry 1999) of Earth repair, which we can *all* help bring to reality.

My conclusion here is that economic growth of the magnitude of the last few (wasteful) decades is both impractical ecologically *and* unethical (if we assume equity and justice for both all people *and* for nonhuman Nature, which should indeed have moral standing). I hope this article has helped to inject some critical thinking into the common assumption that economic growth *must* always grow (and the faster the better). Rather than a focus only on 'sustainable economic growth', I suggest that it is time to focus centrally on an *ecologically sustainable* economy and future.

**Funding:** This research received no external funding.

**Institutional Review Board Statement:** Not applicable.

**Informed Consent Statement:** Not applicable.

**Data Availability Statement:** Not applicable.

**Conflicts of Interest:** The author declares no conflict of interest.

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
