# Peer review of "Questioning the Assumptions, Sustainability and Ethics of Endless Economic Growth"

_jrfm, doi:10.3390/jrfm14100497_

Round 1
Reviewer 1 Report
The article presents a very interesting and current topic. The low environmental and conservation awareness of most of the world's liberal economics is a very serious problem. The ideas provided are clear and very interesting. The author is an expert on the subject and demonstrates his knowledge in the article.
As a recommendation, sometimes many of the ideas that are provided and on which the study is based are self-cited. It would be advisable to support these theories in other authors.
A significant contribution of data that demonstrates what has been explained and thus verify the theories is also missing. I also recommend that these data appear in graphical mode for a better understanding. It would also be advisable to have a final part, before the conclusions, in which the implications that this article may have for both academics and economic sectors are exposed.
Reviewer 2 Report
Many thanks for the possibility to review for JRFM. My observations are summarized below:
- Interesting article, but too theoretical. It seems that it is an article of philosophy than economics or management.
- theoretical considerations about the economy, population, ecology, etc. should be based on analytical research. Must be completed.
- when making only theoretical analysis, indicate the latest literature on the subject (2020/2021) and not from 6-7 years or 11 years ago. Must be improved
- lack of methodology in the article (theses, hypotheses, etc.). Must be completed
However, I don´t think that the manuscript can be in its current version accepted as a lot of work still has to be done. Please carefully restructure the manuscript so that the content of individual section is in line with the expectations for individual sections.
In conclusion , the article lacks the accent of "originality" supported by an in-depth analysis of the literature and / or empirical research (simulations, forecasts).
Author Response
Reviewer 2
Many thanks for the possibility to review for JRFM. My observations are summarized below:
- Interesting article, but too theoretical. It seems that it is an article of philosophy than economics or management.
This is an article that considers environmental science (my main field) plus ecological economics (that I have worked on for some 10 years) and ecological ethics (that I have worked on for 7 years or so). As it includes ecological ethics it does have links to philosophy but cannot be said to be primarily ‘an article of philosophy’. It also extensively refers to ecological economics as well as neoclassical economics. So it is an article with a strong economics content. An extra section on ‘Questioning the assumptions of neoclassical economics’ has now been added to improve such discussion. The topic ‘sustainable economic growth’ of course does have philosophical and ethical aspects that should be (and are in tis article) discussed.
- theoretical considerations about the economy, population, ecology, etc. should be based on analytical research. Must be completed.
Previously, I have referred to the established environmental indicators (ecological footprint, extinction rate, Living Planet Index etc) which summarise decades of research in environmental science. I have now (after the constructive suggestion of Reviewer 1) included two graphs by Prof Will Steffen of ANU that summarise ‘Earth system trends’ and ‘Planetary boundaries’. The data on environmental science and population has been supported by many references (including many several new ones) and is beyond contestation.
- when making only theoretical analysis, indicate the latest literature on the subject (2020/2021) and not from 6-7 years or 11 years ago. Must be improved
This is not just theoretical analysis. The lucidity and rationality of articles does not fade because papers are a few years old. However, another 38 references have now been added that bring some sections up to date in terms of chronology. I would note however that none of them in any way have detracted from (or superceded) the somewhat older (but excellently researched and some of them landmark) references used.
- lack of methodology in the article (theses, hypotheses, etc.). Must be completed
I was asked by JRFM to write this article on the basis of my expertise in this area over three decades of research and writing (as noted by Reviewer 1). The article starts with an introduction and finishes with a conclusion, with 10 sections in between that cover the key topics I wish to discuss, given many threads are involved in this topic. This is not an undergraduate thesis or paper that has to list ‘methods’ ‘results’ and ‘discussion’ but a big picture overview of a topic that involves many different aspects. I have been lucky enough to research and write on these aspects over three decades, and this brings them together to seek to create critical thinking about ‘sustainable economic growth’.
However, I don´t think that the manuscript can be in its current version accepted as a lot of work still has to be done. Please carefully restructure the manuscript so that the content of individual section is in line with the expectations for individual sections.
The content of the individual sections is very much in line with my expectations for these sections, based on three decades of writing about them at a high academic level.
In conclusion , the article lacks the accent of "originality" supported by an in-depth analysis of the literature and / or empirical research (simulations, forecasts).
I appreciate that Reviewer 2 does not agree with the stance of the article, and this is of course her/his right to have this opinion. However, good academic writing seeks to create dialogue that includes differing opinions. A difference of opinion (not fact) should thus not shut down an article. The claim that an article ‘lacks originality’ is often used when a reviewer does not agree with the argument an article is making. However, I note she/he has not been able to suggest any article that does what this article does as a big picture overview of the topic. Clearly it does thus show originality. In fact it fulfills the goal I undertook to the journal, being to provide a big picture overview that questions endless economic growth. This article will at least provide a different lens to what I suspect may be several other articles in the special edition that argue in support of endless economic growth. As journals are meant to encourage dialogue on such important issues, my differing stance can validly be seen as part of such an important dialogue. I thank JRFM for asking me to contribute to this special issue, and in so doing put forward a different analysis than that commonly argued in a large part of academia (that supports endless economic growth). Such differing stances on the issue allow the reader to make up her/his mind.
Round 2
Reviewer 1 Report
Accepted